# Endocrine Follow-Up of Men with Non-Obstructive Azoospermia Following Testicular Sperm Extraction

**DOI:** 10.3390/jcm10153323

**Published:** 2021-07-28

**Authors:** Evangelia Billa, George A. Kanakis, Dimitrios G. Goulis

**Affiliations:** 1Unit of Reproductive Endocrinology, 1st Department of Obstetrics and Gynecology, Medical School, Aristotle University of Thessaloniki, 56403 Thessaloniki, Greece; dgg@auth.gr; 2IVF Unit, Department of Endocrinology, Athens Naval and Veteran Affairs Hospital, 11521 Athens, Greece

**Keywords:** hypogonadism, intracytoplasmic sperm injection, Sertoli cell-only syndrome

## Abstract

Testicular sperm extraction (TESE) is a surgical procedure which, combined with intracytoplasmic sperm injection, constitutes the main treatment for achieving biological parenthood for patients with infertility due to non-obstructive azoospermia (NOA). Although it is effective, TESE procedures might cause structural testicular damage leading to Leydig cell dysfunction and, consequently, temporary or even permanent hypogonadism with long-term health consequences. To a lesser extent, the same complications have been reported for microdissection TESE, which is considered less invasive. The resulting hypogonadism is more profound and of longer duration in patients with Klinefelter syndrome compared with other NOA causes. Most studies on serum follicle-stimulating hormone and luteinizing hormone concentrations negatively correlate with total testosterone concentrations, which depends on the underlying histology. As hypogonadism is usually temporary, and a watchful waiting approach for about 12 months postoperative is suggested. In cases where replacement therapy with testosterone is indicated, temporary discontinuation of treatment may promote the expected recovery of testosterone secretion and revise the decision for long-term treatment.

## 1. Introduction

Testicular sperm extraction (TESE) is a surgical procedure which, in combination with intracytoplasmic sperm injection (ICSI), is currently used to enable men with non-obstructive azoospermia (NOA) to produce their biological children. Several TESE techniques have been reported including simple or multi-biopsy conventional TESE (cTESE), microdissection TESE (micro-TESE, mTESE), and testicular sperm aspiration (TESA) [1]. Their development was imposed by the need for focused, less invasive, and more effective techniques for sperm retrieval, as spermatogenesis is focal in many patients with NOA [2].

The cTESE procedure involves random single or multiple testicular incisions in different testicular regions with a resection of a variable volume of tissue until sperm are identified and extracted [3]. In mTESE, a larger longitudinal or equatorial incision is made through the tunica albuginea under the observation of an operating microscope. The exposed seminiferous tubules are then studied. The larger, more opaque, whitish ones are selectively removed since they are more likely to contain sperm [4]. There is a lack of strong evidence concerning the superiority of one technique over the other in terms of sperm retrieval, pregnancy rates, and live birth rates. The results depend on NOA causes and testicular histology, the latter being a heterogeneous entity with distinct pathological patterns, ranging from hypospermatogenesis to Sertoli cell-only syndrome (SCOS) [5,6,7,8]. However, a recent meta-analysis indicates that mTESE has a 1.5 times higher sperm retrieval rate (SRR) compared with cTESE and 2 times higher rate compared with TESA. Therefore, mTESE should be preferred in men with NOA according to AUA/ASRM guidelines [9,10].

Concerning the postoperative complications, TESE procedures might cause structural testicular damage leading to Leydig cell dysfunction and, consequently, hypogonadism with long-term health consequences [7,11,12,13]. This review aims to discuss the hormonal disturbances after TESE procedures and to suggest an appropriate endocrine follow-up for men with NOA. It will be restricted mainly to hypogonadism due to the lack of evidence for other endocrinological complications.

## 2. Methods

The relevant literature was reviewed through the PubMed, Scopus, and CENTRAL electronic databases to identify the best available evidence concerning endocrine consequences after TESE procedures. All studies and review articles reporting hormonal evaluations and symptoms compatible with hypogonadism before and after TESE were reviewed.

## 3. Complications

Structural and hormonal postoperative complications have been described following TESE procedures [12,13,14]. These complications could be explained by the peculiarities of testicular vascularization and the detrimental effects that the TESE techniques may impose. Testicular blood supply is provided by the testicular artery (a branch of the internal spermatic artery) which enters the testis posteriorly at the level of the mid-pole beneath the epididymis, continues inferiorly to the lower pole, and then ascends along the anterior surface. By this point, it forms sub-capsular arteries, which give rise to centripetal branches that supply the adjacent testicular lobules [15]. Multiple incisions of the tunica albuginea, performed during conventional TESE, may injure sub-capsular vessels and their branches, compromising the blood supply to the corresponding parenchyma or resulting in sub-capsular hematomas. The latter may disrupt testicular function by increasing intratesticular pressure. Impaired blood flow, devascularization, hematomas, and inflammation have been reported one to three months after TESE, leading to testicular scars and calcifications [12,14]. An increase in peritubular scar tissue has also been suggested to affect Leydig cells and germ cells number [16].

mTESE, on the other hand, was initially considered less invasive compared with conventional TESE since the magnified vision minimizes the amount of tissue excised and the risk of inadvertent intra-operative vascular injury [17]. Nevertheless, the same complications have been reported, although to a lesser extent [12,18,19].

These structural disturbances have raised safety issues regarding testicular function. Several men with infertility already have impaired Leydig cell function and, therefore, lower serum testosterone (T), higher luteinizing hormone (LH) and estradiol concentrations compared with fertile men. In addition, many of them have small testicular volumes, which may get further compromised after the extraction of a considerable amount of testicular tissue for sperm retrieval during the TESE procedure [20]. Therefore, they are at high risk of developing androgen deficiency and further disturbance of spermatogenesis and Sertoli cells’ function after TESE [16,21,22].

Despite the limited evidence in this field, there has been a consistent finding of a decline in total testosterone (TT) secretion of variable severity and duration after TESE procedures [8,12]. This decrease of TT concentrations is typically temporary, although two cases of testicular atrophy have been documented [14,23]. The risk of permanent hypogonadism is lesser when smaller testicular samples are taken, as serum TT concentrations return to baseline in 50–90% of patients one year after mTESE [8,11,12,17,24]. The recovery period of hypogonadism that may develop after mTESE is 12–18 months [23,25]. The range in the recovery rates is mostly due to the heterogeneous populations of men with NOA in the various studies. Factors such as the size of the removed testicular tissue, the extension of the tunic incisions, or the experience of the surgeon may also contribute to the recovery rates [8,26].

In a cohort study of highly selected eugonadal men, presenting a low risk of hypogonadism before undergoing bilateral mTESE, the mean postoperative serum TT was 88 ng/dL lower compared with the corresponding preoperative concentrations 19 months after the procedure. Of these men, 30% became biochemically hypogonadal [TT < 300 ng/dL (10.4 nmol/L)] after mTESE [8]. In another study of 435 men with non-obstructive azoospermia who underwent cTESE or mTESE, TT concentrations showed a 20% decline from baseline within three to six months following TESE. In the cTESE group, serum TT concentrations declined from 316 ng/dL to 251 ng/dL while in mTESE group they declined from 303 ng/dL to 248 ng/dL. This decline was more evident in men who underwent more than two attempts of sperm retrieval, while it was not related to the patients’ age, initial TT concentrations, serum FSH concentrations, or the outcome of sperm retrieval (success vs. non-success) [17]. In addition, a study assessing the long-term effects of mTESE in 45 men with NOA (average follow-up: 2.4 years) demonstrated that postoperative T concentrations declined by 10% and that 16% of patients developed a de novo androgen deficiency during follow-up, independently of their age or testicular volume [11]. Furthermore, another study evaluating endocrinological data before and 3, 6, and 12 months after surgery in 69 NOA patients with or without Klinefelter syndrome (KS) showed a significant decrease in TT concentrations to 1.3 ± 0.2 ng/mL (46.4% of the preoperative concentrations) at 6 months after mTESE procedure in men with NOA and 46, XY karyotype. This decline was followed a different pattern according to testicular histology and was greater in patients with 47, XXY karyotype. In patients with hypospermatogenesis, preoperative serum TT concentrations were relatively high (5.2 ± 0.7 ng/mL) and decreased slightly without reaching hypogonadal concentrations. In patients with maturation arrest and SCOS preoperative serum TT concentrations were 2.9 ± 0.2 mg/mL and 3.1 ± 0.3 ng/mL, respectively, and reduced slightly at six months postoperatively to hypogonadal levels (<3.0 ng/mL) [27].

In line with the above data, a recent meta-analysis of 12 non-randomized, retrospective, uncontrolled studies showed a decrease in TT concentrations at 3, 6, 9 and 12 months following TESE procedures compared with baseline results, even reaching hypogonadal levels in some cases [7]. This decrease was more profound in men with KS [7]. In both men with NOA and a 46, XY karyotype and men with KS, the highest decrease in TT concentrations were observed six months after TESE, with a mean decrease of 78 and 118 ng/dL (2.7 and 4.1 nmol/L), respectively, which recovered to baseline at 18 and 26 months, respectively [7]. Limitations of this study include the heterogeneity of the procedures performed, including both obstructive azoospermia (OA) and NOA patients who carry a different risk for hypogonadism and flaws in patient follow-up.

Concerning hormonal disturbances in patients with KS undergoing TESE, the evidence is scarce. According to the above study, preoperative TT concentrations were relatively low (2.8 ± 1.6 ng/mL) and decreased by 30% to 35% at 1 to 12 months postoperatively [24]. In another study, preoperative TT concentrations were 2.8 ± 0.4 ng/mL and decreased significantly after mTESE [27]. A recent meta-analysis also showed decreased TT, with the strongest one being at 6 months after TESE with a mean decrease of 4.13 nmol/L [7].

The severity and duration of the decrease in the serum TT concentrations have been associated with the underlying histology of the seminiferous epithelium. Accordingly, the recovery rates 12 months after the mTESE procedure were complete in patients with hypospermatogenesis (surpassing 100% of the preoperative concentrations) and almost complete in patients with maturation arrest or SCOS (93.1% and 80.6% of the preoperative concentrations, respectively) [27]. This is not the case concerning patients with KS, in whom T concentrations were recovered in only 50% of subjects, 12 months postoperatively. This difference may be attributed to the low testicular volume of patients with KS and the severe histological disorder, which, apart from the seminiferous tubules, affects Leydig cells [24].

Evidence indicates that the prevalence of hypogonadism in men with NOA is up to 45–47% [7,14,22,28]. Men with hypospermatogenesis have normal TT concentrations, while men with maturation arrest and SCOS have lower and sometimes borderline or even low TT concentrations. The latter subgroups are presented with a greater risk of hypogonadism due to further reduction of T after TESE procedures. The reduction of Leydig cells’ number due to tissue removal has been proposed as a possible mechanism of T decrease [7]. The greater the removed tissue and the smaller the preoperative testicle volume, the larger the decline of TT concentrations is expected to reach hypogonadal levels in the subgroup with the lower preoperative levels. The reduction of T concentrations induces LH secretion through the negative feedback mechanism on the hypothalamic-pituitary-testicular axis to stimulate the remaining Leydig cells. The patients that develop post-TESE hypogonadism may have an intrinsic resistance of the cells to the stimulatory effect of LH, since some patients with NOA show an impaired response of Leydig cells in the stimulatory effect of human chorionic gonadotropin (hCG) [7,11,22]. Moreover, a decreased responsiveness of the hypothalamus and pituitary to low T may be another reason for postsurgical hypogonadism [11,27]. Additionally, the recovery of hypogonadism may be related to the Leydig cells’ renewal coming from stem Leydig cells. The time and the degree of renewal may differ in NOA patients due to an underlying pathology of stem cells. Finally, as mentioned above, the inflammation and vascular damage of the testicular parenchyma may lead to Leydig cells dysfunction and/or impair the release of intratesticular T to the circulation. Especially, patients with KS have small testis with solid brown Leydig cell nodules, which may already have a degree of dysfunction responsible for their low T concentrations [29]. These abnormal Leydig cells are more vulnerable to further disruption after a TESE procedure, leading to more profound, prolonged, and even permanent hypogonadism in some cases. Furthermore, it has been implied that it may have a less responsive hypothalamic-pituitary-testicular axis [7,27,30].

Current evidence is inconsistent concerning the TESE consequences on serum follicle-stimulating hormone (FSH) and LH concentrations. Some investigators have reported an increase in the mean serum FSH and LH concentrations in patients with maturation arrest and SCOS after mTESE, while no change was found in patients with hypospermatogenesis [27]. Other studies have documented an increase in serum LH and FSH concentrations in patients with NOA and 46, XY karyotype but not in those with KS [24]. They also suggested that the change in FSH concentrations has resulted from the scar tissue with germ cell loss near the scar after the procedure [24]. In the study mentioned above, the pre-and postoperative FSH and LH concentrations remained low in patients with hypospermatogenesis and high in patients with KS. In patients with maturation arrest, both gonadotropins increased continuously after mTESE, while in patients with SCOS, they increased up to six months after surgery and decreased after that [27].

In a meta-analysis mentioned above, LH concentrations were negatively associated with TT in most studies. A role of LH in the recovery rate has also been implied since men with low TT concentrations had an adequate response, with increased TT concentrations three and four days after hCG injection. In line with this observation, in cases where LH was upregulated immediately following TESE, a faster recovery (18 months) was expected. In one study with KS patients, there was no increase in LH concentrations [24].

Symptoms associated with hypogonadism due to TESE procedures have been described, especially when T concentrations are <12 nmol/L, mainly erectile dysfunction (ED) and decrease of testicular volume [7]. In one study including 66 patients, 13 new-onset erectile dysfunction (ED) cases have been described after mTESE, which significant decreased TT concentrations from 27.1 to 9.7 nmol/L [31]. Some researchers have been reported a reduction of testicular volume of at least 2 mL in 25% of men undergoing cTESE and 2.5% of men undergoing mTESE at six months after the procedure [19]. In addition, another study reported a 0.3 and 0.6 mL decrease in mean testicular volume 3 and 12 months after mTESE [32].

## 4. Endocrine Follow-Up and Treatment

As there is convincing evidence for an increased risk for low TT concentrations following TESE, long-term endocrinological follow-up should be advised in these patients [7,11]. It is important to emphasize that the applied techniques should prevent testicular damage by balancing between tissue sparing and maximization of sperm retrieval rates to avoid repeated procedures. [8]. In line with this approach, some researchers suggest a stepwise strategy. A single TESE sample is initially extracted, followed by an mTESE using the same testicular incision, followed by a multi-biopsy cTESE approach on the opposite testis if needed [33]. Others advocate performing a Fine Needle Aspiration (FNA) mapping before TESE. This technique relies on obtaining 18 testicular aspiration samples to represent the entire testicular surface and depth. According to its supporters, it allows the ensuing TESE to be more focused and less traumatic by avoiding areas with unfavorable cytology [8].

According to the available data, serum TT levels decline in hypogonadal levels (<300 ng/mL) in men with KS and in a proportion of NOA men without KS with the major reduction seeing at six months postoperatively. This reduction is accompanied by a corresponding increase in LH concentrations, implying the possibility and time of recovery. On this ground, evaluation of serum TT and LH concentrations at 3, 6, 12, 18 and, in some cases, 24 months is advisable. As NOA patients are at increased risk of developing hypogonadism, in general, annual estimation of TT concentrations could be, also, considered.

Since T deficiency following TESE is usually temporary, it seems prudent to wait for about 12 months postoperatively until some degree of spontaneous recovery is observed [23]. Nevertheless, some patients with T deficiency may experience symptoms such as erectile dysfunction [7]. In such a case, T replacement therapy (TRT) should be initiated based on a combination of biochemical diagnosis with symptoms of hypogonadism by the current guidelines. Temporary discontinuation of treatment may reveal the expected recovery of T secretion and revise the decision for TRT [26,34]. hCG or selective estrogen receptors modulators (SERMs) administration could be considered in highly selected, hypogonadal patients who have not completed their fertility attempts to increase intratesticular T concentration and manage the hypogonadal symptoms.

## 5. Conclusions

In conclusion, the current evidence suggests a considerable risk of temporary and even permanent hypogonadism following TESE procedures. It is yet unclear whether the increased hypogonadism risk is related to the number or size of testicular tissue samples excised, the number and size of the testicular tunica albuginea incisions, or the surgical experience [8]. Most probably, it is related to the NOA etiology and testicular volume as it is more profound and of longer duration in patients with KS compared with other NOA causes.

## Data Availability

No new data were created or analyzed in this study.

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
