# Peer review of "Endocrine Follow-Up of Men with Non-Obstructive Azoospermia Following Testicular Sperm Extraction"

_jcm, 2021, doi:10.3390/jcm10153323_

Round 1
Reviewer 1 Report
This is a comprehensive review of the literature on the hormonal status of men with non-obstructive azoospermia following testicular sperm extraction. The article gives the information interesting for andrologists taking care of men with NOA. It is correctly written. I have no specific comments.
Author Response
Dear editor
The authors would like to thank the editors and the reviewers for the time and effort in reviewing this manuscript. All reviewers' suggestions and recommendations have been taken under consideration. The changes have been marked up using the “Track Changes” function.
Reviewer: 1
|
Comment |
This is a comprehensive review of the literature on the hormonal status of men with non-obstructive azoospermia following testicular sperm extraction. The article gives the information interesting for andrologists taking care of men with NOA. It is correctly written. I have no specific comments |
|
Response |
We thank the reviewer for his comment. We appreciate it a lot. |
|
Action |
No relative changes have been made. |
|
|
|
Reviewer 2 Report
Billa et al’s manuscript reviews the endocrine related consequences of surgical sperm extraction in men with non-obstructive azoospermia. While the authors bring up several valid points the manuscript would benefit from narrowing its scope and delving deeper into the primary and secondary literature available. While the title suggests this is about multiple endocrinopathies it focuses almost exclusively on testosterone. I would rework the manuscript to talk more about primary and secondary testosterone deficiency and specifically focus on testosterone levels, symptomatology, follow up, testosterone replacement vs. SERM/LH agonist therapy and testosterone recovery in this population.
Abstract:
- I would include a comment about microdissection testicular sperm extraction in the first sentence rather than traditional TESE
- Unclear what the connection between TESE and FSH/LH are in the abstract I would remove altogether and/or expand on this
Introduction:
- The first sentence is copy/pasted from the abstract, consider rewording one
- While I agree with the authors that there is not a one-size-fits-all model for sperm extraction in NOA, a fairly contemporary meta-analysis does indicate that mTESE has a SRR x1.5 of cTESE and x2.0 of TESA
- Bernie et al Fertil Steril 2015 (PMID 26263080)
- This is also stated by the recently minted ASRM guidelines
- Schlegel et al Fertil Steril 2020 PMIDs 33309062, 33309061)
Methods:
- There is no methods section. While this isn’t a systematic review there should be some indication on how articles were selected for review / what was focused on, over what time period, etc.
Complications:
- The authors do a good job of pointing out several complications that may go unrecognized.
- Consider making subsections withing complications
- Consider expanding on the pathophysiology of 3 separate outcomes 1) decrease in TT (pre op / post op) 2) increased risk of low T (<300ng/dL, pre op normal – post op abnormal) 3) increased risk of testosterone deficiency (<300ng/dL and symptoms)
- The last category is missing from the manuscript and has far less primary literature, however should ultimately be the focus of the manuscript. While up to 20% of men may have a 20% decrease in their TT compared to pre-op how many men have symptoms, how many men require treatment, how many men benefit from treatment.
- Consider KS only section to better organize the outcomes
- Consider expanding on the pathophysiology of 3 separate outcomes 1) decrease in TT (pre op / post op) 2) increased risk of low T (<300ng/dL, pre op normal – post op abnormal) 3) increased risk of testosterone deficiency (<300ng/dL and symptoms)
Endocrine follow-up and Treatment:
- This section should discuss when a patient should be followed and with what labs (TT, LH, E?) at what schedule (montly, q3m, etc)
- This section also introduces other variables like FNA mapping, but doesn’t delve into the complication rates with that procedure, cost, etc would recommend discussing in more detail or removing
- This section should also tie in with the above section discussing the merits of hCG and clomid after TESE in attempt to stimulate Leydig cells rather than jump to TRT unless the authors consider those part of the “TRT” armamentarium (just need some clarification here).
Conclusion:
Per the review, at worst an SCO patient has a 20% chance of not recovering baseline function with no comment on the developments of testosterone related symptoms. The review’s conclusion states that there is a “high risk” of “hypogonadism” from TESE procedures. This statement should be revised to better reflect the data presented.
Reviewer 3 Report
Before ms will be further proceed I suggest major improvements:
1. There is only 26 references cited reflecting that Authors probably do not detailly searched available literature
2. Table compering data from various crucial articles referring to hormonal, structural alterations after TESE should be provided or more details should be provided into the text e.g. type of Leydig cell alterations, any Sertoli cell alterations?, concentrations of inhibin, SHBG, etc
3. 3. mTESE, cTESE—please explain
Round 2
Reviewer 3 Report
Authors corrected the paper according suggestions